# Two Degrees of Freedom Synchronous Motion Modulation Technique Using MEMS Voltage-Controlled Oscillator-Based Phase-Locked Loop for Magnetoresistive Sensing

**DOI:** 10.3390/s25061835

**Published:** 2025-03-15

**Authors:** Zhenyu Shi, Zhenxiang Qi, Haoqi Lyu, Qifeng Jiao, Chen Chen, Xudong Zou

**Affiliations:** 1State Key Laboratory of Transducer Technology, Aerospace Information Research Institute, Chinese Academy of Sciences, Beijing 100190, China; shizhenyu17@mails.ucas.ac.cn (Z.S.); qizhenxiang21@mails.ucas.ac.cn (Z.Q.); lvhaoqi19@mails.ucas.ac.cn (H.L.); jiaoqifeng19@mails.ucas.ac.cn (Q.J.); 2School of Electronic, Electrical and Communication Engineering, University of Chinese Academy of Sciences, Beijing 100049, China; chenchen243@mails.ucas.ac.cn; 3QiLu Aerospace Information Research Institute, Jinan 250101, China

**Keywords:** MEMS magnetoresistive sensors, magnetic field motion modulation, phase-locked loop circuit, synchronous technology

## Abstract

This study presents a novel dual phase-locked loop two-dimensional synchronized motion modulation (TDSMM-DPLL) system designed to enhance the low-frequency detection capability of magnetoresistive (MR) sensors by effectively mitigating 1/f noise. The TDSMM-DPLL system integrates a comb-driven resonator and a piezoelectric cantilever beam resonator, achieving synchronized magnetic field modulation through a DPLL circuit that adjusts the resonant frequency of the comb-driven resonator to twice that of the cantilever beam resonator. Theoretical analysis and finite element simulations demonstrate a modulation efficiency of 38.98%, which is significantly higher than that of traditional one-dimensional modulation methods. Experimental validation confirms the system’s effectiveness, showing a 3.13-fold reduction in frequency Allan variance, decreasing from 217.32 ppb to 69.46 ppb, indicating substantial noise suppression. These results highlight the TDSMM-DPLL system’s potential to improve the performance of MR sensors in low-frequency applications, making it a promising solution for high-precision magnetic field detection.

## 1. Introduction

High-precision, low-power, integrated magnetic sensors are in high demand across various fields such as transportation, medicine, military, and aviation [1,2,3,4,5]. In particular, magnetic tunnel junctions (MTJs) with MgO-barriers exhibit exceptionally high magnetoresistance at room temperature and have been extensively utilized in tunnel magnetoresistance (TMR) sensors [6,7]. However, during the magnetization reversal process of the MTJ ferromagnetic layer, 1/f noise is inevitably generated [8], leading to a noise level in the high-frequency range (above 1 kHz) that is nearly two orders of magnitude lower than that in the low-frequency range (10 Hz) [9,10,11,12]. Such 1/f noise severely limits the application of MR sensors, as many practical magnetic signals of interest lie within the low-frequency range (<10 Hz) [13]. Addressing this issue is, therefore, a pressing scientific challenge in the field of spintronic devices. Mitigating 1/f noise and enhancing magnetic field detection capabilities are crucial for unlocking the full potential of MR sensors in critical applications such as medical diagnostics, advanced navigation, and defense technologies.

In 2002, Jander et al. proposed using chopping to reduce 1/f noise in MR sensors. However, their results indicated that the device did not achieve high resolution in the low-frequency range [14]. Edelstein et al. [15,16] proposed utilizing comb teeth and torsion beams to mechanically modulate low-frequency magnetic fields to higher frequencies, effectively mitigating 1/f noise. Similarly, Pan et al. [11] and Hu et al. [17] utilized a piezoelectric beam to induce vibrations above an MR sensor, periodically modulating the magnetic field through a magnetic film deposited on the underside of the piezoelectric cantilever. Subsequently, Guedes et al. [18] reported a vertical motion flux modulation technique, which employs a pair of piezoelectric cantilevers with MFCs to modulate low-frequency magnetic fields, achieving a modulation efficiency of 13% and a resolution of 905 pT/√Hz. Similarly, Tian et al. proposed a vertical magnetic flux modulation strategy, achieving a modulation efficiency of 18.8% and a detection resolution of 0.44 nT/√Hz [19].

Despite these advancements, challenges persist in achieving both high modulation efficiency and high magnetic sensitivity simultaneously. Liu et al. [20] introduced synchronous motion modulation (SMM), which integrates two modulation methods by enabling the piezoelectric cantilever and comb transducer to vibrate synchronously in the vertical and horizontal directions, respectively. Simulation results demonstrated a significant improvement in modulation efficiency, reaching 100.31%. Jiao et al. [21] proposed a two-dimensional synchronized motion modulation structure combined with a comb-driven resonator. Theoretical analysis and simulations suggested that the modulation efficiency of this scheme could reach as high as 127%. Furthermore, Lyu et al. [22] developed a novel MTJ motion modulation (MMM) technology, incorporating MFC fixation and integrating MTJ films grown on the tip of a piezoelectric cantilever that follows the cantilever’s vibration. This approach resulted in an increase in magnetic sensitivity to 2283.3%/mT, a magnetic gain of 39.3, and a magnetic field resolution of 160 pT/√Hz, facilitating the measurement of picotesla-level magnetic fields. In another study, Ma et al. [23] proposed a phase-locked loop with a MEMS voltage-controlled oscillator, which achieved frequency synchronization of two accelerometers through circuit phase adjustment. Wei et al. reported a MEMS Huygens clock based on the synchronization principle, in which a two-channel lock-in amplifier (HF2LI, Zurich Instruments, Zurich, Switzerland) was utilized to achieve phase locking between two MEMS oscillators [24]. This approach provides a valuable reference for our investigation of synchronous modulated motion.

The aforementioned synchronous modulation structures and strategies exhibit high modulation efficiency. However, due to fabrication imperfections, achieving direct synchronization of these devices remains a significant challenge. Existing methods often necessitate complex tuning of the drive circuit, which introduces additional design constraints and increases power consumption. To address this issue, frequency redundancy should be incorporated at the design stage, and synchronization should be achieved through fine-tuning the driving and biasing signals. This approach inevitably increases the complexity of the drive interface circuit for TMR devices. To date, no studies have reported on the design of drive interface circuits for TMR-based synchronous modulation devices.

To address these limitations, this paper proposes a dual phase-locked loop two-dimensional synchronized motion modulation (TDSMM-DPLL) system based on MEMS magnetic sensors. The system features an in-plane two-dimensional synchronized motion modulation (TDSMM) resonator, which integrates a comb resonator and a piezoelectric cantilever beam resonator, along with a dual phase-locked loop drive circuit (DPLL). The in-plane comb-driven resonator leverages the spring softening characteristics of the parallel plate [25,26] and is tuned to function as the MEMS VCO of the system through electrostatic negative stiffness tuning. The resonant frequency of the comb-driven resonator is adjusted to be twice that of the piezoelectric cantilever beam resonator by varying the DC bias voltage, which is generated based on the phase difference between the two resonators. Once the system stabilizes, the two resonators operate synchronously with a fixed frequency difference, allowing for the modulation of the low-frequency magnetic field at twice the resonant frequency of the piezoelectric cantilever beam. The newly developed TDSMM-DPLL system not only provides a practical synchronous drive circuit but also introduces an innovative magnetic field modulation scheme. This innovation significantly enhances modulation efficiency and holds potential for superior magnetic field detection resolution. Furthermore, this innovative dual-resonator approach offers a promising solution for achieving synchronized modulation with reduced noise [24], offering significant potential for MEMS-based magnetic field sensing applications.

This paper is structured as follows: Section 2 presents the operating principle and theoretical analysis of the TDSMM-DPLL, along with the corresponding finite element simulation results. Section 3 reports the experimental validation, while Section 4 summarizes the findings and discusses potential future research directions.

## 2. Materials and Methods

### 2.1. Overview

The DPLL consists of eight functional blocks: resonator, TIA, phase detector (PD), filter, adder, frequency divider, MEMS VCO, and phase shifter, as illustrated in Figure 1. The MEMS VCO is driven by a phase-locked loop (PLL) circuit. The output signal generated by the MEMS VCO passes through the frequency divider before driving Oscillator 2. The PD compares the phase difference between the output signal and the reference signal to generate an error voltage that is proportional to this phase difference. The LPF then filters the error voltage signal to produce a DC voltage. An adder adds the DC voltage with the bias voltage required for Resonator 1, thereby generating the necessary bias voltage for that resonator. Together, the phase detector, loop filter, and MEMS VCO form a dual phase-locked loop system.

As illustrated in Figure 1, the DPLL consists of eight functional blocks: resonator, Transimpedance Amplifier (TIA), phase detector (PD), filter, adder, frequency divider, MEMS Voltage-Controlled Oscillator (MEMS VCO), and phase shifter. The MEMS VCO serves as Oscillator 1, which is driven by a phase-locked loop (PLL) circuit. The output signal generated by the MEMS VCO passes through the frequency divider before driving Oscillator 2. The PD compares the phase difference between the output signal and the reference signal, generating an error voltage that is proportional to this phase difference. This error voltage is then filtered by the low-pass filter (LPF) to produce a DC voltage (***v***). The adder combines ***v*** with the bias voltage ($V_{0}$) supplied by the circuit, thereby generating the appropriate bias voltage ($V_{dc1}$) for Oscillator 1. Collectively, the phase detector, loop filter, and MEMSVCO collectively constitute a dual phase-locked loop system.

The system functions correctly only when specific conditions are met. According to the dynamic pull-in theory of phase-locked loops [27], the system achieves locking when the resonant frequency of Resonator 1 falls within the capture range of the phase-locked loop. Therefore, for normal operation, the following conditions must be satisfied: (1) the total phase shift in the loop must equal zero; (2) the inherent frequencies of the two resonators must be sufficiently close, or a mechanism must be in place to align their frequencies. The first condition can be addressed by utilizing the phase shifter module within the circuit. The second condition, however, must be fulfilled through careful circuit design. Since the resonant frequency of a resonator is influenced by its bias voltage, the output frequency of Resonator 1 can be adjusted by varying its bias voltage. In the MEMS VCO, the resonant frequency is tracked via a phase-locked loop [28,29]. By modifying the control voltage of the MEMS VCO, the frequencies of the two resonators can be aligned, ultimately achieving synchronization [23]. To minimize the need for excessively high bias voltage (Vdc1), it is advisable to select resonators with inherently similar frequencies.

### 2.2. MEMS-VCO

Figure 2a illustrates the structure of the TDSMM-based MR resonator. Two magnetic flux concentrators (MFCs) are deposited on two lateral silicon masses, which serve to concentrate the magnetic field spatially. The two moving masses are connected to six fixed guide beams. The MTJ, used for sensing the applied magnetic field, is positioned at the top of the cantilever beam, with the six surrounding beams forming a double-ended tuning fork (DETF). To synchronize the two MFCs, the DETF couples them into a unified system, compelling them to vibrate at their resonant frequency. For MTJs, only thermal noise is significant when the resonant frequency exceeds the corner frequency of the typical 1/f noise spectrum, which is generally around 10 kHz.

The DETF-based comb-driven resonator consists of six individual beams and various movable components. Three beams, connected to the left moving mass, form one tine of the DETF, while the other three beams, connected to the right moving mass, constitute the second tine of the DETF. By modeling the DETF comb structure presented in this paper, we obtain a two-degree-of-freedom system, the governing equation of which can be expressed as follows [30]:(1)$${\ddot{x}}_{1}+c\dot{x_{1}}+kx_{1}+c_{12}\left(\dot{x_{1}}-\dot{x_{2}}\right)+k_{12}\left(x_{1}-x_{2}\right)=0$$(2)$$m{\ddot{x}}_{2}+c\dot{x_{2}}+kx_{2}+c_{12}\left(\dot{x_{2}}-\dot{x_{1}}\right)+k_{12}\left(x_{2}-x_{1}\right)=0$$
where ***m*** represents the individual effective lumped masses of each tine, ***k*** denotes the individual effective lumped spring constants of each tine, ***c*** is the individual effective lumped damping coefficient acting on each tine, and $c_{12}$ and $k_{12}$ are the coupling damping and stiffness coefficients, respectively.

To achieve higher modulation efficiency and improved low-frequency magnetic field resolution, the comb-drive MEMS resonator based on the DETF designed in this paper operates in an out-of-phase mode [31]. Additionally, when the DETF vibrates in an out-of-phase mode within a vacuum environment, the damping effects can be neglected, and the stress in the connecting beam can be assumed to be completely offset. Therefore, the out-of-phase angular frequency can be derived by solving the above equation as follows:(3)$$\omega_{0}^{2}=\frac{k}{m}$$

When the resonator vibrates in an out-of-phase mode, there is no coupling effect between the left and right comb teeth, indicating that the entire system is effectively connected to six separate fixed-guided beams. Based on existing literature regarding the mechanical lumped parameters of a single fixed-guided beam [31], the equivalent mechanical lumped parameters of the system can be derived as follows:(4)$$M_{eff}=\frac{39}{35}\rho AL+m_{tot},\;k_{linear}=\frac{36EI}{L^{3}},\;k_{nonlinear}=\frac{54EA}{25L^{3}}$$
where ***L*** is the length of the beam, ***E*** is the Young’s modulus, ***ρ*** is the density, ***A*** is the cross-sectional area, ***I*** is the moment of inertia, $m_{tot}$ is the total mass of the two moving masses, and $M_{eff}$, $k_{linear}$, and $k_{nolinear}$ are the effective mass, linear spring constant, and nonlinear spring constant of the entire system, respectively.

The schematic illustrating the electrical connections of the driving port and sensing port is presented in Figure 2b,c. The driving ports are supplied with an alternating current (AC) voltage, denoted as $v_{ac}$, while the movable part of the resonator is biased by a direct current (DC) voltage, $V_{dc}$. The sensing port is linked to an external transimpedance amplifier (TIA). The system parameters are defined as follows: $x_{0}$ represents the initial overlap distance, *g* is the spacing between the fixed and moving fingers, *d* corresponds to the initial separation between the fixed finger and the moving mass, while $w_{f}$ denote the height of the finger, respectively. The total electrostatic force exerted on the resonator can be categorized into two contributions: one arising from the driving ports and the other from the sensing ports.

When the comb-tooth resonator is driven to vibrate, the electrostatic force from the longitudinal capacitance can be expressed as follows:(5)$$F_{d,l}=\frac{1}{2}\frac{\partial C_{d,l}}{\partial x}{\left(V_{dc}+v_{ac}\right)}^{2}=\frac{1}{2}\frac{\partial C_{d,l}}{\partial x}\left(V_{dc}^{2}+2V_{dc}v_{ac}\right)\left(v_{ac}\ll V_{dc}\right)$$(6)$$\frac{\partial C_{d,l}}{\partial x}=\frac{\partial }{\partial x}\left[\frac{2N_{d}\varepsilon \left(x_{0}+x\right)h}{g}\right]=\frac{2N_{d}\varepsilon h}{g}=\alpha$$

Similarly, the electrostatic force from the transverse capacitance can be expressed as follows:(7)$$F_{d,t}=\frac{1}{2}\frac{\partial C_{d,t}}{\partial x}{\left(V_{dc}+v_{ac}\right)}^{2}=\frac{1}{2}\frac{\partial C_{d,t}}{\partial x}\left(V_{dc}^{2}+2V_{dc}v_{ac}\right)\left(v_{ac}\ll V_{dc}\right)$$(8)$$\frac{\partial C_{d,t}}{\partial x}=\frac{\partial }{\partial x}\left[\frac{\left(2N_{d}+1\right)sw_{f}h}{\left(d-x\right)}\right]=\frac{\left(2N_{d}+1\right)\varepsilon w_{f}h}{(d-x{)}^{2}}=\frac{\beta }{(d-x{)}^{2}}$$
where $V_{dc}$ represents the DC bias voltage, $v_{ac}$ represents the AC drive voltage, ***ε*** is the permittivity of air, ***h*** is thickness of fingers, $N_{d}$ is the number of fingers in the driving port (with $N_{d}$ = 38), and $C_{d,l}$ and $C_{d,t}$ represent the longitudinal and transverse capacitances at the driving end, respectively. Therefore, the total electrostatic force at the driving end can then be expressed as follows:(9)$$F_{d}=F_{d,l}+F_{d,t}=\frac{1}{2}[\alpha +\frac{\beta }{{\left(d-x\right)}^{2}}](V_{dc}^{2}+2V_{dc}v_{ac})$$

The electrostatic force at the detection end can be easily obtained as follows:(10)$$F_{s}=\frac{1}{2}\frac{\partial C_{d,z}}{\partial z}V_{dc}^{2}=\frac{1}{2}\frac{\varepsilon S}{{\left(d_{0}-z\right)}^{2}}V_{dc}^{2}=\frac{1}{2}\frac{\gamma }{{\left(d_{0}-x\right)}^{2}}V_{dc}^{2}$$
where ***S*** is the area of the capacitance and $c_{d,z}$ is the capacitance at the detection end.

The total electrostatic force can be expressed as follows:(11)$$F_{tot}=F_{d}+F_{s}=\frac{1}{2}\left[\alpha +\frac{\beta }{(d-x{)}^{2}}\right]\left(V_{dc}^{2}+2V_{dc}v_{ac}\right)+\frac{1}{2}\frac{\gamma }{({d_{0}-x)}^{2}}V_{dc}^{2}$$

Taking damping into account, the governing equation for the comb-drive resonator is given as follows:(12)$$M_{\mathrm{eff}}\ddot{x}+C_{\mathrm{tot}}\dot{x}+k_{\mathrm{linear}}x+k_{\mathrm{nonlinear}}x^{3}=F_{\mathrm{tot}}$$
where $c_{tot}$ is the damping coefficient including all damping, using the Taylor expansion, and the governing equation can be rewritten as follows:(13)$$M_{eff}\ddot{x}+C_{tot}+\left(k_{linear}-k_{e1}\right)x+k_{e2}x^{2}+\left(k_{nonlinear}-k_{e3}\right)x^{3}\;=\left(\frac{1}{2}\alpha +\frac{\beta }{d^{2}}\right)\left(V_{dc}^{2}+2V_{dc}v_{dc}\right)+\frac{\gamma }{d_{0}^{2}}V_{dc}^{2}$$
(14)$$k_{e1}=\frac{2\beta }{d^{3}}\left(V_{dc}^{2}+2V_{dc}V_{ac}\right)+\frac{2\gamma }{{d_{0}}^{3}}V_{dc}^{2},\;k_{e2}=\frac{3\beta }{d^{4}}\left(V_{dc}^{2}+2V_{dc}V_{ac}\right)+\frac{4\gamma }{{d_{0}}^{4}}V_{dc}^{2},\;k_{e3}=\frac{4\beta }{d^{5}}\left(V_{dc}^{2}+2V_{dc}V_{ac}\right)+\frac{4\gamma }{{d_{0}}^{5}}V_{dc}^{2}$$
where $k_{e1}$, $k_{e2}$, and $k_{e3}$ are the first-order, second-order, and third-order electrostatic spring constant.

Substituting Equation (14) into Equation (3) yields the following equation:(15)$$\omega_{0}=\sqrt{\frac{\left(\frac{36EI}{L^{3}}-{K\left(V_{dc}+{mV}_{ac}\right)}^{2}+{nV}_{ac}^{2}\right)}{\frac{39}{35}\rho AL+m_{tot}}}$$
where ***m***, ***n*** and ***K*** represent the coefficients of the quadratic equation formula for $V_{dc}$. No further mathematical derivation is done here. From Equation (15), it is evident that the anti-phase mode resonant frequency of the comb-drive resonator is negatively correlated with the bias voltage. Consequently, the resonant frequency can be adjusted by varying the bias voltage.

When a bias voltage is applied, the phase of the switching circuit can be synchronized with the natural frequency of the comb-driven resonator through a phase-locked loop. The frequency of the resonator can be adjusted by varying the bias voltage to ensure it remains within the capture range of the phase-locked loop, while also satisfying the loop’s phase condition to maintain synchronization. In this configuration, the comb-driven resonator and the phase-locked loop together form a MEMS VCO. By applying an appropriate bias voltage, the comb-driven resonator can be tuned to operate at a specific frequency.

### 2.3. System Model

The simplified structure of the TDSMM-DPLL system is shown in Figure 3. The phase detector (PD) detects the phase difference between the input voltage ***cosθ1*** and the reference voltage ***cosθ***, producing a voltage signal ***u***, which is then filtered by a low-pass filter to generate a DC voltage signal ***v***. The MEMS VCO modifies the system’s resonant frequency through electrostatic negative stiffness when the AC excitation of the comb drive remains fixed. $k_{v}$ is defined as the electrostatic stiffness coefficient, and the output of the MEMS VCO can be obtained accordingly:(16)$$\dot{\theta }=\sqrt{\omega_{v}^{2}-k_{v}{\left(V_{0}+v\right)}^{2}}$$
where $V_{0}$ is the initial bias voltage, and $\omega_{v}$ and $k_{v}$ are defined as follows:(17)$$\omega_{v}^{2}=\frac{\frac{36EI}{L^{3}}+{nV}_{ac}^{2}}{\frac{39}{35}\rho AL+m_{tot}},\;k_{v}=\frac{K}{\frac{39}{35}\rho AL+m_{tot}}$$

Therefore, the transfer function of the system can be expressed as follows:(18)$$H\left(s\right)=\frac{F(s)k_{g}\left(\theta -\theta 1\right)k_{v}}{s}$$
where F(s) is the transfer function of the low-pass filter, and $k_{g}$ represents the gain of the phase detector.

### 2.4. Theoretical Analysis and Simulation of TDSMM

To quantitatively analyze the magnetic field modulation efficiency of the system after synchronization, the modulation efficiencies of two modulation modes are first examined: the magnetically coupled resonator (MMM) driven by the piezoelectric cantilever [22] and the magnetically coupled resonator (MFCMM) driven by the comb [21].

For MMM, the magnetic field configuration (MFC) remains constant, while the magnetically coupled tunnel junction (MTJ) is located at the tip of the piezoelectric cantilever and moves in accordance with its oscillations, resonating at a higher frequency. The correlation between the magnetic field gain and the MTJ position for MMM modulation is illustrated in Figure 4a. The fitting formula is expressed as follows:(19)$$G_{MMM}=m+ne^{-ph^{2}}$$
where ***m***, ***n***, and ***p*** are constants that depend on the material and size of the MFC. Assuming that the DC magnetic field before modulation is $B_{0}$, the magnetic field after modulation can be expressed as follows:(20)$${B_{ac}=B}_{0}\cdot G_{MMM}$$

Furthermore, assuming the resonant frequency is $f_{1}$, and the amplitude is $x_{1}$, the height of the MTJ during motion can be expressed as follows:(21)$$h=x_{1}\sin\left(2\pi f_{1}t\right)$$

By substituting Equation (21) into Equation (20), we obtain the following equation:(22)$${B_{ac1}=B}_{0}\left(m+ne^{-p{(x_{1}\sin\left(2\pi f_{1}t\right))}^{2}}\right)$$

Since the displacement is in the micrometer range, the exponent of the exponential function is much smaller than one. Therefore, Equation (22) can be expanded using a Taylor series as follows:(23)$$B_{ac1}=B_{0}[m+n-\frac{npx_{1}^{2}}{2}(1-\cos\left(2\pi \cdot 2f_{1}t\right))]$$

The DC component of the magnetic field after modulation can then be expressed as follows: ${B_{m,dc1}=B}_{0}\left(m+n-\frac{npx_{1}^{2}}{2}\right)$. The amplitude of the fundamental frequency of the magnetic field after modulation is given by the following equation:${B_{m,ac1}=B}_{0}\frac{npx_{1}^{2}}{2}$. Modulation efficiency is defined as the ratio of the base frequency amplitude to the DC component. Therefore, the modulation efficiency of MMM can be expressed as follows:(24)$$E_{MMM}=\frac{B_{m,ac1}}{B_{m,dc1}}=\frac{\frac{npx_{1}^{2}}{2}}{m+n-\frac{npx_{1}^{2}}{2}}\approx \frac{npx_{1}^{2}}{2(m+n)}$$

From Equation (24), it can be observed that the modulation efficiency is influenced by the material, size, and displacement of the MFC. Additionally, a greater displacement results in higher modulation efficiency.

For MFCMM, the magnetic film at the end of the piezoelectric cantilever remains fixed, while the MFC resonates in the horizontal direction. Assuming that the initial gap of the MFC is $g_{0}$, the resonant frequency is $f_{2}$ and the amplitude is $x_{2}$. Figure 4b illustrates the relationship between the magnetic field gain and the MFC position in MFCMM modulation. The fitting formula is provided as follows:(25)$$G_{MFCMM}=a+be^{-cg}$$
where $g=g_{0}-2x_{2}sin\left(2\pi f_{2}t\right)$, which indicates that the MFCs move closer to each other initially and then move apart after reaching their closest position. Using the same analytical approach, the modulation efficiency of GCM can be expressed as follows:(26)$$B_{m,ac2}=2B_{0}bcx_{2}e^{-cg_{0}}sin\left(2\pi f_{2}t\right)$$(27)$$E_{MFCMM}=\frac{B_{m,ac2}}{B_{m,dc2}}=\frac{2cx_{2}}{1+a/b^{e}e^{cg_{0}}}$$
where ***a***, ***b***, and ***c*** are constants related to the material and geometric parameters of the MFC.

To further enhance the modulation efficiency of MR-MEMS sensors, this paper explores the combination of two modulation methods and proposes in-plane two-dimensional synchronized motion modulation (TDSMM). As shown in Figure 2a, there is no direct coupling between the comb-drive resonator and the piezoelectric cantilever beam resonator. Therefore, to simplify the analysis, we define the magnetic field gain of TDSMM as the product of the gains from MFCMM and MMM as follows:$$G_{TDSMM}=G_{MMM}\cdot G_{MFCMM}=\left(m+ne^{-ph^{2}}\right)\cdot \left(a+be^{-cg}\right)$$(28)$$=\left(a+be^{-cg_{0}}\right)\left(m+n-\frac{npx_{1}^{2}}{2}\right)+\left(a+be^{-cg_{0}}\right)\frac{npx_{1}^{2}}{2}\cos\left(2\pi \cdot 2f_{1}t\right)\;+2bcx_{2}e^{-cg_{0}}\left(m+n-\frac{npx_{1}^{2}}{2}\right)\sin\left(2\pi f_{2}t\right)$$

From Equation (23), it can be seen that the magnetic field frequency after MMM modulation is twice the resonant frequency of the cantilever beam resonator. From Equation (26), it is clear that the magnetic field frequency after MFCMM modulation matches the frequency of the comb-driven resonator. Therefore, for TDSMM modulation, to maintain a single magnetic field frequency after modulation, the resonant frequency of the comb-driven resonator must be twice that of the cantilever beam resonator, i.e., $f_{2}=2f_{1}$. Therefore, the amplitudes of the DC and AC components of the TDSMM are expressed as follows:(29)$$G_{TDSMM,0}=\left(a+be^{-cg_{0}}\right)\left(m+n-\frac{npx_{1}^{2}}{2}\right)$$(30)$$G_{TDSMM,1}=\left(a+be^{-cg_{0}}\right)\frac{npx_{1}^{2}}{2}+2bcx_{2}e^{-cg_{0}}\left(m+n-\frac{npx_{1}^{2}}{2}\right)$$

Therefore, the modulation efficiency of SMM can then be expressed as follows:(31)$$E_{TDSMM}=\frac{G_{TDSMM,1}}{G_{TDSMM,0}}=\frac{px_{1}}{m/ne^{pH_{0}-1}}+\frac{2cx_{2}}{1+a/b^{e^{cg_{0}}}}=E_{VMM}+E_{GCM}$$

From Equation (31), it can be observed that the efficiency of TDSMM is simply the sum of the modulation efficiencies of MFCMM and MMM. This indicates that, for the same MTJ, the MEMS magnetoresistive sensor based on SMM achieves a higher magnetic field measurement resolution. Furthermore, the parameters influencing MFCMM and MMM similarly affect TDSMM, meaning that the modulation efficiency of TDSMM increases with the amplitude of the resonator. To calculate the modulation efficiency of two-dimensional motion modulation, we present an alternative formula for the modulation efficiency:(32)$$E=\frac{B_{max}-B_{min}}{{2B}_{s}}\times 100\%$$

### 2.5. Mechanical Structure

The cross-sectional view of the four-mask fabrication process for the presented device is illustrated in Figure 5a. The device employs a TFoS (Thin Film on Silicon) structure on an SOI (Silicon On Insulator) substrate, which includes a one-micron-thick aluminum nitride film and a 10-micron-thick silicon layer. The SOI device layer is highly doped, enabling the direct growth of the piezoelectric material on the silicon without the necessity for a bottom metal electrode. The fabrication process begins with the deposition of a piezoelectric thin film on the front side of the substrate. Silicon dioxide (SiO_₂_) is then grown and patterned using PECVD (Plasma-Enhanced Chemical Vapor Deposition) as a mask. The aluminum nitride is etched with a 25% TMAH (Tetramethylammonium Hydroxide) solution, followed by the removal of the SiO_₂_ mask. Next, a layer of titanium/gold is evaporated and subjected to a lift-off process to form the top electrodes and pads for wire bonding. Deep reactive ion etching (DRIE) is used to pattern the SOI layer for device structures and electrostatic gaps. Finally, the handle layer and buried oxide layer are etched from the backside to release the device [31,32,33]. Figure 5b presents an SEM image of the fabricated resonator. Figure 5c illustrates the shape of the piezoelectric cantilever beam resonator along with the electrode routing. Figure 5d provides a close-up view of the comb-driven parallel plate detection, featuring an electrode gap of approximately 4 μm. Table 1 lists the key parameters of the demonstrated resonators.

## 3. Results

The experimental measurements were conducted under vacuum conditions using a mechanical pump, a molecular pump, and a vacuum chamber, which maintained an internal pressure of 2.9 × 10^-4^ Pa inside the chamber. The designed circuit and its connection scheme are illustrated in Figure 6. Specifically, Figure 6a shows the PCB circuit designed for this study, Figure 6b presents the testing environment for measuring the electrical properties of the circuit, Figure 6c depicts the setup for testing the resonator’s oscillation displacement, and Figure 6d provides a microscopic view of the resonator’s electrical connections within the vacuum cavity during displacement testing.

Figure 7 displays the simulated and measured resonant frequency results of the fabricated TDSMM resonator. The observed discrepancies in resonant frequency can be attributed to fabrication imperfections. Figure 7a illustrates the scanning results of the cantilever beam, which shows a resonant frequency of approximately 15.45 kHz and a Q factor ranging from 8000 to 9000. Figure 7b presents the scanning results of the comb-drive resonator under a bias voltage of 80 V, with a resonant frequency of approximately 30.97 kHz. Notably, the resonant frequency of the comb-drive resonator is nearly twice that of the cantilever beam resonator, thereby satisfying the conditions for closed-loop synchronization.

Figure 8a shows the results of a single closed-loop scan for the MEMS-VCO, which consists of the inner loop of the designed circuit and the comb-tooth resonator, under the same excitation and different bias voltages. To achieve the required driving voltage during the experiment, the ATA-4315 high-voltage power amplifier from Aigtek was used to amplify the external loop frequency difference voltage signal, reaching the voltage value that satisfies the closed-loop conditions. Figure 8b shows the voltage–frequency relationship and the corresponding fitting results. From these results, we can derive Equation (33), where $k_{v}=9.45\times 10^{4}H_{z}^{2}/V^{2}$. (33)$$f^{2}=1.02\times 10^{9}-9.45\times 10^{4}\;V^{2}$$

Figure 9a illustrates the time-domain waveform of the driving signal for the two resonators when the comb-drive resonator and the cantilever beam resonator are synchronized under the operation of the dual closed-loop circuit. At this stage, the bias voltage of the comb-drive resonator is approximately 81.568 V. Resonator 1 refers to the comb-drive resonator, while Resonator 2 corresponds to the cantilever beam resonator. As observed in the figure, the resonant frequency of the comb-drive resonator is twice that of the cantilever beam resonator when synchronized.

By incorporating the measured displacement of the two resonators (as shown in Figure 10) into the finite element simulation model, the time-domain variation of the magnetic field gain for the three modulation modes—MFCMM, MMM, and TDSMM—can be obtained, as illustrated in Figure 9b TDSMM exhibits a higher peak-to-peak value compared to MFCMM and MRLMM, demonstrating the highest modulation efficiency. The simulation results indicate that when the resonance frequency of MFCMM modulation is twice that of MMM modulation, a single-frequency modulated magnetic field can be achieved.

Figure 11 illustrates the Allan variance of the frequency output from the inner ring comb-drive resonator, both before and after the system achieves closed-loop synchronization. The frequency meter system output was recorded over a duration of 2 h. Prior to closed-loop synchronization, the frequency output exhibited a bias instability of 217.32 ppb with an average time of 3.32 s. After closed-loop synchronization, the bias instability was reduced to 69.46 ppb with an averaging time of 7.07 s, representing a 3.13-fold improvement. These results demonstrate that the dual closed-loop synchronization circuit not only effectively synchronizes the two resonators but also enhances the motion performance of the resonators during closed-loop operation.

To characterize the displacement of the resonator, both the cantilever beam resonator and the comb-driven resonator were measured using the Polytec Laser Vibrometer MSA-600 (Polytec, Waldbronn, Germany). The displacement of the cantilever beam corresponds to vertical motion, which can be easily characterized by the laser. The time-domain measurement results for the cantilever beam resonator’s displacement are shown in Figure 11a, with a maximum displacement of 40 µm in the vertical plane. The displacement of the comb-driven resonator is an in-plane displacement. The measurement principle involves characterizing the relative displacement at the measurement point by comparing the coordinate differences of two framed areas with significant contrast at different resonator phases. The displacement measurement results for the comb-driven resonator are shown in Figure 11b, with a maximum displacement of 500 nm in the in-plane direction.

In the simulation, $B_{0}=10,000\;nT$. The simulation results indicate that the DC component of the magnetic field after modulation is $B_{s}=56.299\;\mu T$. Using the modulation efficiency Formula (32), the modulation efficiency of TDSMM modulation can be calculated as follows: $E_{TDSMM}=38.98\%\;$ (where $E_{MFCMM}=3.45\%,\;E_{MMM}=37.21\%$). Due to the omission of high-order harmonic terms, the combined modulation efficiencies of the two methods are slightly higher than that of TDSMM modulation. Table 2 compares the modulation efficiencies of various MR-MEMS sensors, demonstrating that the modulation efficiency of two-dimensional motion modulation is significantly higher than that of one-dimensional modulation. This comparison highlights the feasibility of the proposed two-dimensional motion modulation system, representing a significant advancement toward achieving an effective and practical solution for low-frequency magnetic field sensing.

Although the modulation efficiency of the method proposed in this paper is lower than that of SMM and TDSMM (x-y), the results obtained from iterating the displacement of the resonator after synchronization are closer to the actual values compared to the simulated results of SMM and TDSMM (x-y). This indicates that the circuit presented in this paper successfully achieves two-dimensional synchronous motion of the resonator, effectively addressing the challenges associated with implementing a two-dimensional motion modulation circuit. The system’s robustness and precision for low-frequency magnetic field sensing are now firmly established, demonstrating its potential as a reliable and scalable solution.

While the small amplitude of the designed comb resonator, as illustrated in Figure 9b and Figure 11b, significantly affects the modulation efficiency of the MFCMM, a judicious combination of one-dimensional modulation schemes can further enhance the modulation efficiency of the MR-MEMS sensor. This has theoretical implications for optimizing the modulation structure. Future research will concentrate on refining the resonator’s design to increase its displacement amplitude, as well as improving the integration of the drive circuit to ensure better scalability and performance under various environmental conditions. Considering that the resonant frequency of the designed TDSMM resonator is at the level of 30 kHz, the design and implementation of a low-frequency phase-locked loop will be a major challenge for ASIC integration This work lays a promising foundation for the development of miniaturized, highly sensitive, and robust magnetic field sensors with broad applicability in both military and industrial sensing applications. There may be some possible limitations in this study. Specifically, the experimental implementation of the TDSMM sensor was confined to vacuum chamber environments, potentially omitting the effects of environmental parameters such as temperature on resonator performance characteristics. Following the completion of all microfabrication stages for the resonator, vacuum encapsulation will be performed, after which variable–temperature experiments will be conducted under controlled conditions to quantitatively assess how temperature fluctuations affect the magnetic modulation performance of the sensor. Based on the experimental results, temperature compensation measures will be further optimized, including the implementation of a temperature compensation circuit to enhance frequency stability and structural optimizations in the resonator design to minimize thermal expansion effects.

## 4. Conclusions

In this study, we propose a two-dimensional synchronized motion modulation (TDSMM) system based on a dual phase-locked loop (DPLL) for magnetoresistive (MR) sensors, aimed at mitigating the significant 1/f noise interference in MR elements. A batch of TDSMM resonators, fabricated using the SOI process, has been successfully developed. Through theoretical analysis, finite element simulations, and experimental validation, we demonstrate the feasibility of the TDSMM-DPLL system for synchronized magnetic field modulation.

The synchronization of the two resonators—the comb-driven resonator and the piezoelectric cantilever beam resonator—was successfully achieved using the DPLL circuit, confirming the viability of the synchronous modulation approach. Experimental results indicated that the resonant frequency of the comb-driven resonator was approximately twice that of the cantilever beam resonator, thereby fulfilling the necessary conditions for closed-loop synchronization.

The modulation efficiency of the TDSMM system was theoretically analyzed and simulated, resulting in a modulation efficiency of approximately 38.98%. This value closely aligns with the combined efficiencies of the two individual modulation techniques, MFCMM and MMM, and is significantly higher than that of one-dimensional modulation approaches. Experimental results further demonstrated that the DPLL circuit effectively improved the resonator’s noise characteristics, aiding in the suppression of noise in the measured magnetic field signal. Allan variance analysis revealed a substantial reduction in bias instability: the frequency bias instability of the inner-loop comb-driven resonator decreased from 217.32 ppb to 69.46 ppb, representing a 3.13-fold improvement. This underscores the effectiveness of the DPLL circuit in enhancing resonator performance during closed-loop operation.

Future work will concentrate on optimizing the structure of MEMS-MR sensors and developing a fully integrated magnetic field measurement chip. This chip will co-integrate an ASIC with the MEMS-MR unit, aiming to achieve higher modulation efficiency and reduced noise levels. This research not only offers an innovative solution for low-frequency magnetic field detection using MR sensors but also establishes a foundation for the development of high-performance magnetic sensors in the future.

## Figures and Tables

**Figure 1 sensors-25-01835-f001:**
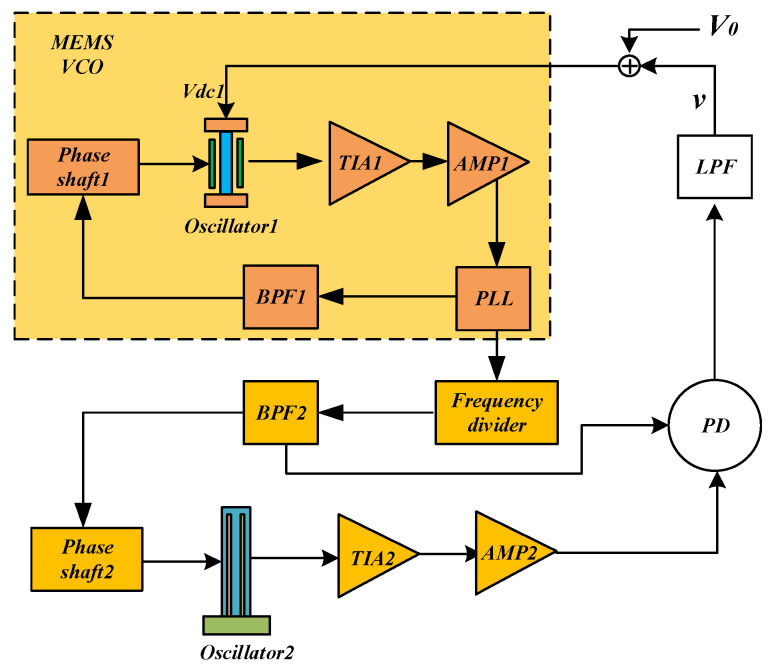
Block diagram of the overall system of DPLL.

**Figure 2 sensors-25-01835-f002:**
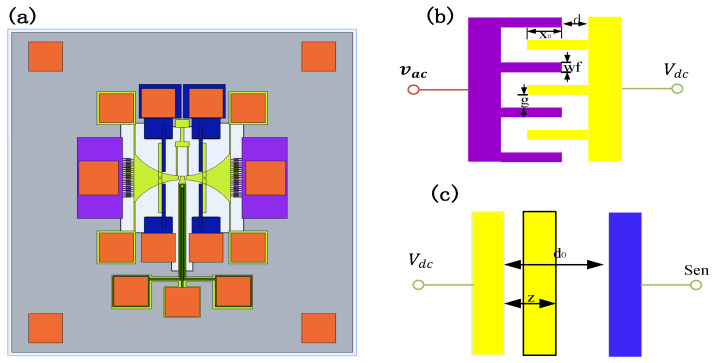
MEMS resonator based on the TDSMM, the purple domain represents the driving terminal, the yellow domain represents the moving part, the orange and blue domains represent the sensing terminal. (**a**) The designed three-dimensional structure; (**b**) the electrical connection of the driving terminal; (**c**) the electrical connection of the sensing terminal.

**Figure 3 sensors-25-01835-f003:**
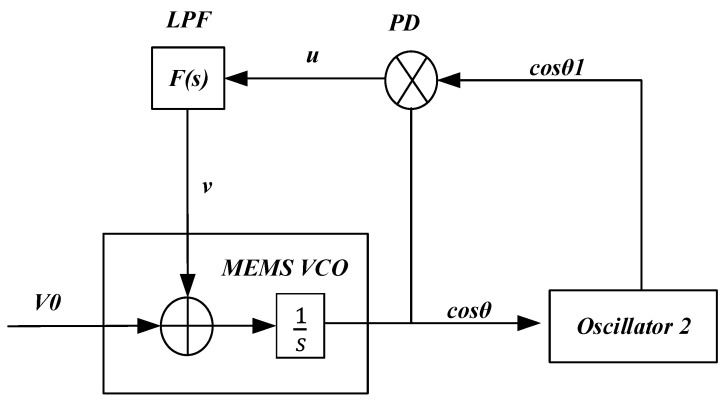
Schematic of TDSMM-DPLL (dual phase-locked loop two-dimensional synchronized motion modulation).

**Figure 4 sensors-25-01835-f004:**
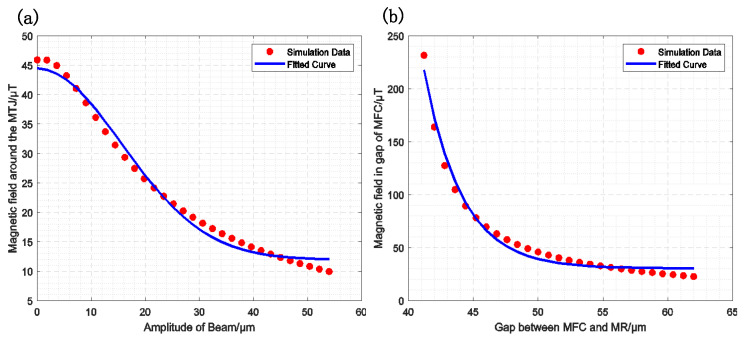
Magnetic field gain simulation fitting curve, the red points represent the simulation results, and the blue curve represents the fitting results. (**a**) The relationship between the MMM-modulated magnetic field gain and the cantilever beam amplitude ($G_{MMM}=m+ne^{-ph^{2}}$); (**b**) The relationship between the MFCMM-modulated magnetic field gain and the gap size ($G_{MFCMM}=a+be^{-cg}$).

**Figure 5 sensors-25-01835-f005:**
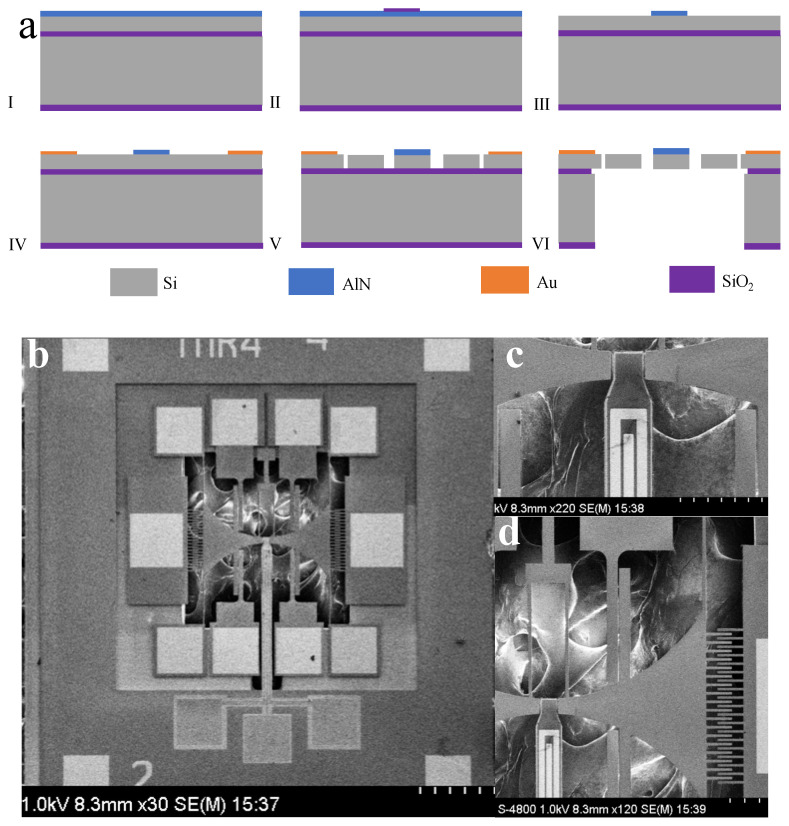
(**a**) Fabrication process flow of TDSMM; (**b**) SEM image of TDSMM chip; (**c**) SEM image of cantilever beam; (**d**) SEM image of drive comb fingers and detect parallel plates.

**Figure 6 sensors-25-01835-f006:**
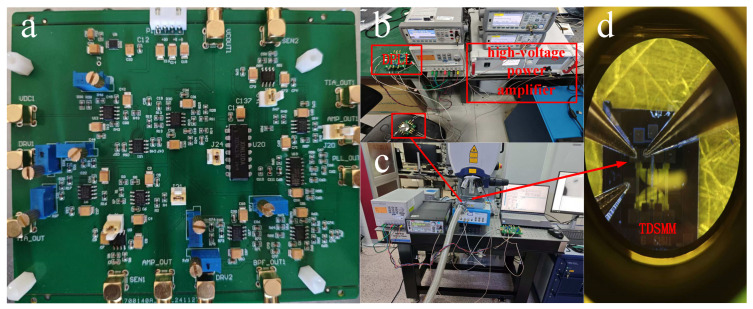
(**a**) DPLL circuit PCB (utilizing a −5 V to 20 V power supply, which provides a bias voltage of up to 19 V); (**b**) Circuit performance testing environment (during testing, the time-domain and frequency-domain characteristics of the two resonators are measured using an oscilloscope and two frequency counters, ensuring that the resonators achieve synchronization; a high-voltage power amplifier is employed to amplify the bias voltage supplied by the circuit, driving the resonators into synchronization). (**c**) Resonator amplitude testing environment (during testing, the lens of the laser vibrometer emits a laser beam that passes through a glass window of the vacuum chamber to scan the resonator). (**d**) Electrical connections within the vacuum chamber.

**Figure 7 sensors-25-01835-f007:**
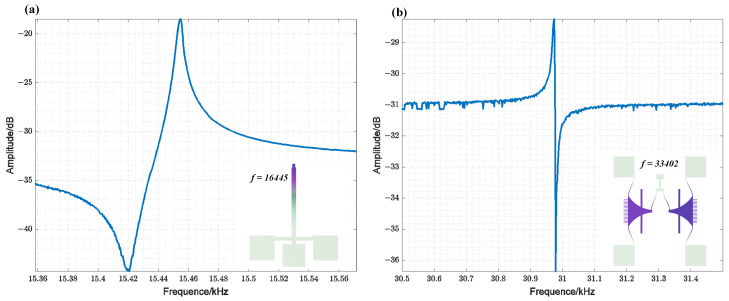
(**a**) Amplitude Frequency Response of the Cantilever Beam Resonator ($v_{ac}\;$ = 100 mV) (insert: modal simulation results of the cantilever beam resonator, with a designed resonant frequency of 16,445 Hz); (**b**) Amplitude Frequency Response of the comb-drive resonator ($V_{dc}$ = 80 V, $v_{ac}$ = 40 mV) (insert: modal simulation results of the comb-drive resonator, with a designed resonant frequency of 33,402 Hz).

**Figure 8 sensors-25-01835-f008:**
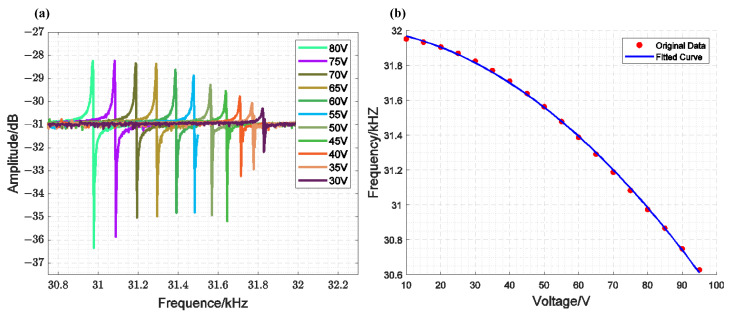
Measurements of the Voltage–Frequency Response of a Single Closed-Loop MEMS-VCO: (**a**) Scanning Results; (**b**) Fitting Curve of the Relationship Between Frequency and Voltage.

**Figure 9 sensors-25-01835-f009:**
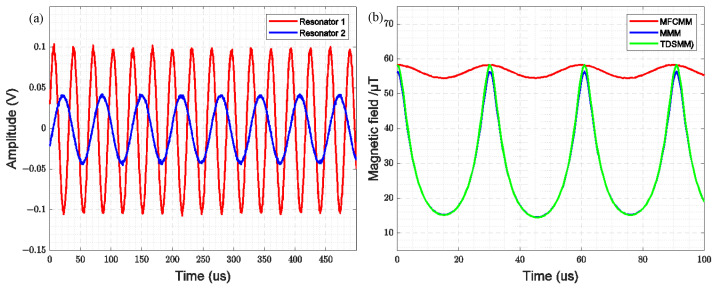
(**a**) Time-Domain Waveform of the Synchronous State; (**b**) Modulation magnetic fields of three modulation methods.

**Figure 10 sensors-25-01835-f010:**
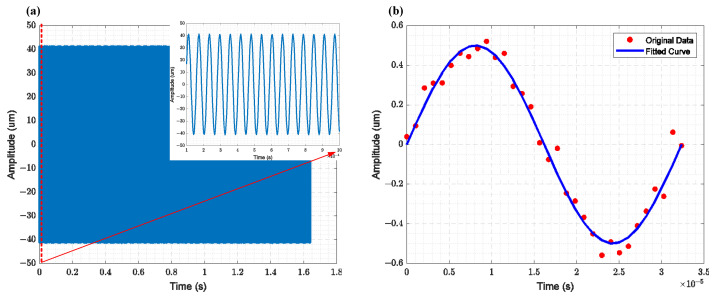
Amplitude measurement results of the resonator obtained using a laser vibrometer: (**a**) The relationship between the longitudinal displacement of the cantilever beam and time (The subfigure displays the measurement results for 100 µs), (**b**) The variation in the lateral displacement of the comb resonator over one cycle, where the red points represent the measured results and the blue line represents the fitted results.

**Figure 11 sensors-25-01835-f011:**
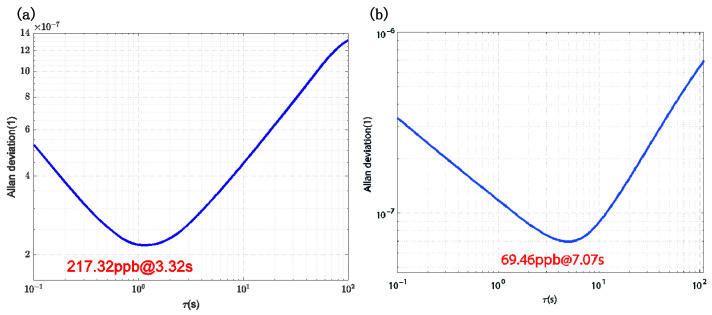
Allan Variance of the Inner Loop: (**a**) Before Synchronization, (**b**) After Synchronization.

**Table 1 sensors-25-01835-t001:** The key parameters of TDSMM.

Parameter	Value [μm]
AIN thickness	1
Length of beam	820
Width of beam (upper)	45
Width of beam (nether)	70
Length of DETF beam	300
Length of short side of transverse mass	40
Length of long side of transverse mass	490
Length of long side of transverse mass	390
Gap of transverse mass	50
Gap of parallel plate	3

**Table 2 sensors-25-01835-t002:** Comparison of modulation efficiency of various MR-MEMS sensors.

Modulation	Types of MR Components	Modulation Efficiency (%)
VMM [18]	GMR	13
MFVMM [19]	MTJ	18.8
MMM [22]	MTJ	8.69
SMM [20]	MTJ	100.31
TDSMM (x-y) [21]	GMR	127
TDSMM (x-z) (this work)	MTJ	38.98

## Data Availability

Data are contained within the article.

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
