# Peer review of "Two Degrees of Freedom Synchronous Motion Modulation Technique Using MEMS Voltage-Controlled Oscillator-Based Phase-Locked Loop for Magnetoresistive Sensing"

_sensors, 2025, doi:10.3390/s25061835_

Round 1
Reviewer 1 Report
Comments and Suggestions for Authors
The paper presents a two-dimensional synchronous motion modulation system (TDSMM) based on a dual phase-locked loop (DPLL) for magnetoresistive sensors. While the TDSMM-DPLL approach is not entirely new, as similar modulation techniques have been explored in previous works, the authors have conducted a comprehensive study through theoretical analysis, FEM simulations, and experimental validation. It is a well-structured and thorough study. However, one key requirement for improvement is that the novelty of this work and its comparison to prior studies should be clearly articulated to highlight its unique contributions.
Reviewer 2 Report
Comments and Suggestions for Authors
The authors present a study on a novel, two-dimensional synchronous motion modulation system with a dual PLL for reducing flicker noise in magnetoresistive sensing. The system combines a comb-driven resonator with a piezoelectric acuated cantilever beam, controlled by a PLL circuit. In my opinion, the paper is very well written, with a solid presentation on theoretical analysis, simulations, and experimental results. I thank the authors for their contribution to this area of research and for submitting a high quality article to Sensors.
After reviewing, I believe a few areas could improve from further clarification:
1. The fab process of the MTJ is missing. It is unclear whether the MTJ used is an MgO tunnel barrier and nor specific materials were used for the ferromagnetic layers. More information on the fabrication and integration steps would be useful. It would be interesting to the broader audience on the address any fabrication challenges related to impurities or defects that could impact performance.
2. The modulation efficiency of 39% is from simulations, while Allan variance measurements demonstrate a ~3x improvement in frequency stability after synchronization. However, it is unclear whether the modulation efficiency was or can be experimentally verified, as the results only confirm frequency stability. A comparison between theoretical predictions and experimental data would provide better insight to the systems performance.
4. The experiments were conducted in a vacuum while most real world applications operate in air. It would be useful to discuss whether atomsphere or other environmental factors would affect performance outside a controlled lab setting.
5. The authors commented on the potential for ASIC integration, but it does not discuss the specific challenges related to circuit design and signal processing. It would be great for the authors to expand on this subject, for example, whether this approach could be adapted for other types of sensors, such as accelerometers or gyroscopes.
Reviewer 3 Report
Comments and Suggestions for Authors
The manuscript under review presents a novel Dual Phase-Locked Loop Two-Dimensional Synchronized Motion Modulation (TDSMM-DPLL) system designed to enhance the low-frequency detection capability of magnetoresistive sensors. The proposed TDSMM-DPLL system achieves synchronized modulation of the magnetic field through the DPLL circuit, and achieves a two-fold increase in the resonant frequency of the comb-driven resonator. Theoretical analysis and simulation, as well as experimental verification, confirm the effectiveness of the system, indicating its high practical significance. However, the article has a number of comments:
- The caption of Figure 7 is on another page, which is unacceptable. There is no reference to Figure 9 (b) in the text.
- The contribution of the proposed approach is not clear in the introduction.
- Too many self-citations. I suggest partially replacing them with articles by other authors.
-
In the article the authors operate a TDSMM method based on two comb-driven resonators. However, the authors do not provide comparisons with other methods, such as magnetic flux concentrators motion modulation, and magnetoresistive elements longitudinal motion modulation methods.
Round 2
Reviewer 1 Report
Comments and Suggestions for Authors
No further comments.